# Crests, troughs, and plateaus: Using story theory to explore the experiences of adolescents and young people living with HIV in Kampala, Uganda

Derrick Nuwahereza[1], Allen Nabisere[1], Imelda Namatovu [1], Tom Denis Ngabirano[1], Charles Peter Osingada [2]*

**1** Department of Nursing, School of Health Sciences, Makerere University College of Health Sciences, Kampala, Uganda, **2** School of Nursing, University of Minnesota, Weaver Densford Hall, Minneapolis, Minnesota, United States of America

* Osing004@umn.edu

## Abstract

### Introduction

Adolescents and young people living with HIV experience significant challenges, including unmet psychosocial and self-management needs and limited access to adequate HIV information. Peer support strategies that allow individuals to share their personal experiences and life stories have shown promise in improving their engagement in care and addressing the psychological and social challenges of living with HIV. However, current care approaches do not optimize individual life stories as a foundation for delivering patient-centered and individualized care.

### Methods

This study applied a story path approach, an innovative way of exploring the experiences of adolescents and young people living with HIV. Specifically, it aimed to identify the high points, low points, and turning points in the health journeys of adolescents and young people living with HIV. This descriptive qualitative study was conducted in 2024. Data were collected through face-to-face individual interviews with adolescents and young people living with HIV and analyzed manually.

### Results

Fifteen participants, aged 15–24 years (average age 21.4, SD 2.4), were interviewed. The findings revealed that participants experienced uplifting from social support provided by friends, family, school authorities, and peers. Despite this support, many participants experienced emotional distress, faced challenges with medication adherence, and struggled with suicidal ideation. Counseling was a vital turning point in helping nearly all participants improve their health trajectories.

**Data availability statement:** All relevant data are within the manuscript and its Supporting Information files.

**Funding:** The author(s) received no specific funding for this work.

**Competing interests:** The authors have declared that no competing interests exist.

**Abbreviations:** AYPLHI: Adolescents and young people Living with HIV, CATS: Community Adolescent Treatment Supporters, HIV: Human Immunodeficiency Virus, TASO: The AIDS Support Organization, UAC: Uganda AIDS Commission.

## Conclusions

Story theory provides a valuable framework for understanding the health experiences of adolescents and young people living with HIV and for delivering care that is tailored to their unique individual narratives. These findings may be useful to nurses, counselors, and peer supporters involved in the care of this population. Future research should explore ways to integrate story theory into adolescent HIV care services to improve health outcomes for adolescents and young people living with HIV.

## Introduction

By the end of 2023, approximately 40 million people were living with HIV globally, and another 630,000 individuals had died of AIDS-related illnesses [1]. East and Southern Africa continue to bear the highest burden of the disease, with approximately 20.8 million people living with HIV [1]. Although Sub-Saharan Africa has registered the biggest overall decline in new HIV infections and AIDS-related deaths [2], adolescent girls and young women in East and Southern Africa remain disproportionately affected by the disease [1]. Besides having a high incidence of HIV, adolescents and young people living with HIV (AYPLHIV) face significant challenges, including unmet psychosocial and self-management needs, as well as limited access to adequate information about HIV.

In Uganda, by the end of 2022, approximately 1.4 million people were living with HIV, including 166,000 adolescents and young people aged 15–24 years [3]. Although nationally there has been a significant decline in new infections and AIDS-related deaths, the incidence of HIV among young Ugandans remains alarmingly high. The government of Uganda prioritizes innovative approaches to reach AYPLHIV with comprehensive prevention, care, and treatment services [4]. Currently in most HIV care and treatment facilities in Uganda, the Young People and Adolescent Peer Supporters (YAPS) model [4] is being implemented. Adapted from Zimbabwe's Community Adolescent Treatment Supporters (CATS) trial [5], the YAPS model leverages young people and adolescents as key service providers to screen, test, and link their peers to care. According to a recent Uganda Aids Commission (UAC) report, the model has resulted in increased case identification, linkage to care, retention in treatment, and improved viral load suppression among AYPLHIV [4].

The success of the YAPS model and similar peer support programs [6] is partly credited to the ability of peer supporters to share their experiences and life stories, positioning themselves as credible sources of information and role models for AYPLHIV [7]. The impact of the YAPS model can be enhanced by weaving intentional approaches that encourage peer supporters and healthcare providers to actively engage in dialogue with their patients to understand their personal narratives. The story theory provides a useful framework to guide such a dialogue. A core concept of Story Theory is intentional dialogue, in which the nurse purposefully engages with the patient to understand the story of the health challenge that is complicating their daily

life [8]. As individuals share their stories, they experience a deeper connection with themselves, leading to an energizing release as the meaning of their story becomes clearer and more coherent [8]. When peer supporters and healthcare providers listen to the stories of young people living with HIV, they foster a deeper understanding of the challenges they encounter and facilitate individualized care. This patient-centered approach helps young individuals feel seen, valued, and more willing to engage during medical visits. Additionally, the understanding that develops allows providers to generate care plans that address both the clinical and psychosocial needs of AYPLHIV, thereby improving their physical and mental health.

To date there has been insufficient exploration of the potential within the personal narratives of young people living with HIV to shape patient-centered care. Limited understanding of individual patient's story hampers the implementation of targeted interventions and support systems. Personal stories reveal the intricate pathways individuals navigate, shedding light on their unique encounters with stigma, and struggles with mental health and social relationships. Training young people and adolescent peer supporters to elicit and engage with these personal stories helps them to gain deeper insights into the psychosocial challenges faced by AYPLHIV. This enables them to deliver more personalized, and efficacious interventions aimed at enhancing the overall well-being of AYPLHIV.

The objective of this study was to applying story theory [9] to explore the experiences of adolescents and young people living with HIV. Specifically, the study focused on identifying the high points, low points, and turning points in the journeys of AYPLHIV. By collecting and analyzing their story paths and narratives, the investigators aimed to understand the complexities of the experiences of AYPLHIV and gain a comprehensive understanding of the challenges they face. Insights reported in this study may help nurses, peer supporters, and other health providers in designing targeted interventions and providing anticipatory counseling tailored to the unique circumstances of these young individuals. Additionally, findings from this study may function to elevate the utility of story theory in the care of AYPLHIV.

## Materials and methods

### Study design and setting

This was a descriptive qualitative study. This design was suitable as it allowed investigators to present straight descriptions of the experiences of participants and staying close to the data without engaging in deep interpretations [10]. This study was conducted at the AIDs Support Organization (TASO) Centre, located in Mulago. The center provides services for patients living within a 75 km radius reach and its catchment area covers the districts of Kampala, Mityana, Mukono, and Wakiso. TASO Mulago provides HIV/AIDS clinical and psychological services to children, adolescents, youths, and adults living with HIV/AIDS. Examples of the services include adherence counseling, condom education and distribution, health education, adherence measurement, tuberculosis screening, opportunistic infections examinations, and laboratory investigations. The clinic days for AYPLHIV are Tuesday, Wednesday, and Thursday. On clinic days, about 60–80 patients come for ART refills and other HIV/AIDS services.

### Study participants

The study was conducted among AYPLHIV aged 15–24 years, that were receiving care and treatment services from TASO. To be included in the study, prospective participants had to be aged 15–24 years, have lived with HIV for at least six months, able to speak English or Luganda, receiving care and treatment from the TASO-Uganda Mulago branch, able to provide informed consent or assent, and willing to participate. Patients who were unable to communicate verbally were excluded. Participants were purposefully identified from those attending adolescent-friendly care services in the study site. Participants were identified with the help of health personnel in the clinics.

## Data collection

Data collection took place from 30th July to 7th August 2024. One time one-on-one in person interviews were conducted with consenting participants at the health facility by I.N. The interviews were conducted in a private room at the study site that offered sufficient auditory privacy, and counseling services were readily available in case any participant experienced emotional distress during the interview. The interview guide used during the interview was developed using guidance provided by the developers of the story theory [9]. The interview guide was piloted in a different health facility from the one where data were collected, and adjustments were made based on the feedback from the pre-test. The interview guide contained items that invited to reflect on their present and past experiences, describe their future hopes and dreams. The interviewer listened to each participant's story to identify the high, low, and turning points in their health experience. Each interview was audio recorded after obtaining permission from the participant. The interview time ranged from 15 to 44 minutes. Additionally, each participant was requested to recall their experiences from the time they got to know their HIV positive status. Each participant was requested to map these experiences on a plain piece of paper provided by the interviewer. Specifically, they were instructed to mark the high points – times when things were going well, low points- when things were not going well, and turning points – deciding moments or shifts in perspective that altered life direction [8]. Before ending the interview, the interviewer looked over the story path created by each participant to ensure accuracy of what had been written. After the interview, each participant was paid 5,000 Uganda shillings (about 1.3 US dollars) to compensate for their time. At the beginning of the study, the investigators had planned to recruit 15–20 participants. However, after conducting 15 interviews, data collection was stopped because no new experiences were emerging from the interviews, indicating that a point of data saturation had been attained [11]. One participant dropped off from the study due personal reasons. This paper reports findings from analysis of story paths drawn by 15 participants.

## Data management and analysis

The 11 English audio recordings and four Luganda recordings were transcribed and translated by I.N. and A.N., who are fluent in both languages and have extensive experience in qualitative research methods. Each transcript was cross-checked against the original recording to ensure accuracy and completeness. To gain an overall understanding of the data, OCP read each of transcripts. OCP has extensive experience in the design, conduct, and reporting of qualitative research and has completed graduate-level coursework in qualitative research. OCP applied a deductive coding approach to manually identify the high points, low points, and turning points in the health trajectories of adolescents and young people living with HIV (AYPLHIV). During coding, these key points were highlighted using different colors to distinguish them. Once all transcripts were coded, the codes were grouped into categories based on whether they represented high, low, or turning points. In addition, each individual story path was manually analyzed to further identify these points. A summary of the findings, including the identified high, low, and turning points, is presented in the results section, accompanied by illustrative quotes. However, it is important to note that although we made efforts to bracket our own perceptions during the interviews, data analysis, and reporting of the findings, the investigators' perspectives may still have inadvertently influenced participants' responses during the one-on-one interviews, as well as the analysis and reporting processes.

## Ethics approval

Ethics approval was obtained from the Makerere University School of Health Sciences Research Ethics Committee (MAKSHSREC-2024–679), and administrative clearance was obtained from the management of TASO Mulago branch. In addition, written informed consent was obtained from the participants before they participated in the study. Furthermore, assent was sought from participants who were less than 18 years old, and their parents or guardians were requested to provide permission for their child to be involved in the study.

## Results

Fifteen participants were interviewed, with ages ranging from 15 to 24 years. The average age was 21.4 years, with a standard deviation of 2.4 years. Nearly three-quarters of the participants were single females. Ten participants (67%) had been living with HIV since birth. Additional sociodemographic characteristics are presented in Table 1.

### High points

Generally, moments of social and emotional support played a crucial role, with participants frequently mentioning the encouragement and care received from family members, health providers, school personnel, and friends as uplifting.

*"We had a doctor here she was called… she also helped me in times when I felt like why am I taking this medicine, she used to call me every day she used to follow me up, in fact she used to call me for those first 2 weeks she used to call me every day to ask how you taken the medicine? how are you feeling now? She even used to give some ka money. For the first weeks when I came back, she told me now you see you are fine the viral load is going low, she told me next time".* **(Participant 5; Male, 21 years)**

*"…when I was well after the sickness, they brought me here I started taking drugs then they took me to my brothers place …to gain confidence, he used to counsel me, …he used to take me to places like forest park those interesting places and he told me everything is going to be okay nothing is going to change apart from just taking drugs… those two months made me feel confident I realized that everything is okay I will be fine then I let go, I had to accept the situation".* **(Participant 5; Male, 21 years)**

*"…she always advised me and whenever she's seeing that... people discussed about HIV things and then maybe sometimes the discussion will just come as one of the disease people are learning in biology. But you know, you feel as if they are talking about you, this kind of thing. She would know that it is touching me. Sometimes she comes after the lesson and says, are you okay? Are you fine? She would want to find out how you are. Yeah. And then also during time for drugs… she's the one who was now reminding me. If we are in prep time and we are still revising, and she has seen*

**Table 1. Demographic characteristics of study participants.**

| Participant Number | Age | Marital status | Sex | Current level of education | Duration lived with HIV |
|---|---|---|---|---|---|
| 1 | 22 | Single | Female | Secondary | 13 months |
| 2 | 21 | Married | Female | Secondary | Since birth |
| 3 | 23 | Single | Female | Primary | Since birth |
| 4 | 24 | Single | Female | University | Since birth |
| 5 | 21 | Single | Male | University | 60 months |
| 6 | 15 | Single | Female | Primary | Since birth |
| 7 | 23 | Single | Female | University | Since birth |
| 8 | 19 | Single | Male | Secondary | Since birth |
| 9 | 22 | Married | Female | Secondary | 36 months |
| 10 | 22 | Single | Female | University | Since birth |
| 11 | 18 | Single | Male | Secondary | Since birth |
| 12 | 24 | Single | Female | University | Since birth |
| 13 | 22 | Married | Female | Primary | 9 months |
| 14 | 21 | Married | Female | University | 84 months |
| 15 | 24 | Single | Male | University | Since birth |

*it's 9 o'clock. She will tap me. She'll come and say, can we go? She will not tell anyone where we are going. She just holds my hand and moves out, as if we are going maybe for a short call or to discuss something. Yet she's reminding me, she wants me to go and take my drugs. Then she comes back and sits in the class". (Participant 12, Female, 24 years)*

In addition, securing a source of income, disclosing HIV status, and receiving positive reinforcement from loved ones also emerged as meaningful high points in the experiences of adolescent and young people living with HIV.

*"…having friends who encourage you to continue doing what you are doing because i have a friend who is older, we met in school, he stays in...(name of location omitted) when we talk, he tells me to continue taking my medicine. He gives me hope". (Participant 8; Male, 19 years)*.

### Low points

Most of participants experienced low moments upon discovering their HIV-positive status. Many struggled with adherence to medication due to stigma, forgetfulness, or the challenges posed by boarding school environments. Experiences of rejection, whether from friends, romantic partners, or the workplace, were also commonly reported, illustrating the pervasive impact of stigma and discrimination.

*"The challenges I have faced there is a lady I used to work with, she kept telling my customers that am HIV positive I felt really bad because they started discriminating me, disrespecting me but I left that place where I did not feel welcomed that is the only challenge I have faced". (Participant 3, Female, 23 years)*

*"Automatically bad, I became emotional, I stigmatized myself, I hated everyone around me, that's how it was". (Participant 4, Female, 24 years)*

Additionally, several participants expressed suicidal ideation, feelings of hopelessness, and self-isolation, underscoring the severe psychological toll associated with HIV diagnosis. Falling sick with illnesses related or unrelated to HIV was also a common low point for this population of adolescents as it increased their fears and worries of losing their lives.

*"I actually felt like committing suicide, it was too much for me and the person that gave it to me actually knew that was the worst thing for me to bear but I went through therapy. I had like 3months of therapy so I just had to accept my condition and move on…. I was broken and I was like did this person not know he was sick. So, I don't know (crying). I went through therapy, I was so depressed I couldn't open up to anyone, couldn't open up to my mum so i told a friend and a friend convinced me to at least tell my mother". (Participant 1, Female, 22 years)*

*"…of course (sighs) hmmn hmmn, I felt like it was over for me, I felt like the world had ended, eeh and I felt like even killing myself it was really a bad situation cause I was totally sick at that time the body was so soft, it had made me go down so I felt like I won't even get well even if I take tablets but later on I was counselled and got some help". (Participant 5; Male, 21 years)*

### Turning points

The turning points reported by participants predominantly revolve around key moments of social support, counseling, and personal acceptance of their HIV-positive status. Many participants identified receiving counseling as a pivotal moment in their lives, helping them cope with their diagnosis, adhere to medication, and manage emotional distress.

*"…I was feeling bad they discriminated me, I was not feeling okay even I wanted to change the school yea but later on our sister those nuns they were like no you do this they counselled me and the nurse was there for me and some of my friends because there were some 2 girls who were also swallowing [taking antiretroviral medicines], they were there for me and I picked up. It wasn't easy….the bigger part was counselling because I used to go to the dormitory and cry after crying, I sleep yea like that. Even if you tried to concentrate and read it wasn't easy. So I used to cry sleep, cry sleep"* **(Participant 7; Female, 23 years)**

*"When the counselor spoke to me and gave me hope that I can still live with it and told me, you just continue with your medication, like do not abandon yourself from people, you continue speaking to them because there is nothing, like if you take your medication, nothing really happens and maybe if... anything has gone wrong, come back and speak to me. If you cannot tell your friend, come back and open up to me"* **(Participant 14; Female, 21 years)**

Acceptance of HIV status and disclosure to supportive individuals, including family members, friends, and peers living with HIV, also played a crucial role in fostering resilience and instilling hope.

*"I accepted it, acceptance was the major thing for me and when I interact with some of my friends that I met here that are HIV positive, it gave me a relief because they have this thing of its just, just take your medication and move on just take the tablet and the rest is okay and the fact that I don't have any other health problem apart from it yea. So, they made it simpler for me, taking my medication and accepting to live with it, only the two things that I have to follow"* **(Participant 1, Female, 22 years)**

*"What matters most, the first thing was acceptance that one gave me hard time but…it came after some good time because even I finished my candidate classes when am in day school when I had not felt like I can keep my own medicine at school. I felt like aah aah acceptance came when I was in A level that's when I accepted, I was like aahh now come what may am not going backwards it has come and it will never go".* **(Participant 5; Male, 21 years)**

## Discussion

This study utilized story theory to explore the experiences of adolescents and young people living with HIV (AYPLHIV). Specifically, we sought to document the high, low, and turning points in their health trajectories. Our findings revealed that the most frequently mentioned high points were receiving encouragement and care from family members, school personnel, and peer networks. Disclosure of HIV status and receiving positive reinforcement from loved ones also emerged as a commonly cited high point. Conversely, most participants experienced their lowest moments upon discovering their HIV-positive status. Additional low points included struggles with medication adherence due to stigma, forgetfulness, and the challenges posed by boarding school environments. Experiences of rejection from friends, romantic partners, and workplaces were also reported. Moreover, several participants expressed suicidal ideation, feelings of hopelessness, and self-isolation, underscoring the severe psychological toll associated with an HIV diagnosis.

Turning points in participants' health trajectories predominantly revolved around key moments of social support, counseling, and personal acceptance of their HIV-positive status. Many participants identified receiving counseling as a pivotal moment, helping them cope with their diagnosis, adhere to medication, and manage emotional distress. Disclosure to supportive individuals, including family members, friends, and peers living with HIV, also played a crucial role in shifting the trajectory of their health journeys.

Our findings indicate that most participants experienced uplifting moments when they received social support from friends, family members, school personnel, and peers. This aligns with existing literature on the importance of social and peer support for AYPLHIV. For example, Willis et al. (2019) found that adolescents who received support from community adolescent treatment supporters experienced improved psychosocial well-being compared to those who did not receive

such support [12]. Additionally, social and peer support have been linked to better medication adherence [13], improved adherence efficacy [14], viral load suppression [15,16] and reduced depression [17]. Path stories analyzed in this study indicated that almost all participants experienced a trough period soon after learning their HIV status. This highlights the importance of linking adolescents and young people to social support services immediately following diagnosis. Early linkage to these services may mitigate the magnitude of psychosocial distress and enhance the potential for a quicker recovery.

Our study also revealed that AYPLHIV frequently experienced low points characterized by nonadherence to medication due to fear of stigma, as well as high levels of suicidal ideation, depression, and hopelessness. These findings are consistent with previous reports documenting the prevalence and spectrum of mental health challenges faced by AYPLHIV [18–21]. While multiple studies have established the mental health burden among this population, few have provided insight into the specific timing of these challenges. By using graphical approaches such as story paths, healthcare providers may be better equipped to offer anticipatory care tailored to the evolving needs of AYPLHIV.

For most adolescents interviewed, receiving counseling was a pivotal turning point, facilitating coping with their diagnosis, adherence to medication, and the management of emotional distress. This finding underscores the importance of HIV counseling services as a critical mechanism for promoting behavior change and psychological well-being [22,23]. The story paths analyzed in this study depict health trajectories marked by crests, troughs, and plateaus, highlighting the need for ongoing counseling and sustained support particularly during the low periods to help AYPLHIV lead meaningful and fulfilling lives.

Our study had a few limitations that should be considered when interpreting the findings. First, we relied on participants' ability to recall their experiences from the time they first learned of their HIV-positive status. Given that some participants had been living with the disease for more than seven years, their recollection of past events may not have been entirely accurate, and this together with our inability to conduct member checking may have affected the quality of the findings. However, the use of a visual mapping approach may have helped participants organize their memories more clearly, which likely improved their ability to identify key experiences they might have otherwise overlooked. Second, despite efforts to balance participant gender, males were underrepresented in our study. This limited our ability to fully explore and capture gender-specific variations in the experiences of adolescents and young people living with HIV (AYPLHIV). Finally, some participants, particularly those with only a primary level of education, struggled with drawing story paths and required support from the research assistants.

## Conclusion

Story theory provides a valuable framework for understanding the health experiences of adolescents and young people living with HIV and for delivering care that is tailored to their unique individual narratives. Using this middle-range theory, our study revealed that AYPLHIV benefit greatly from social support provided by friends, family, school authorities, and peers. At the same time, many experienced emotional distresses after learning their HIV-positive status, faced challenges with medication adherence, and, in some cases, struggled with suicidal ideation. Counseling emerged as a crucial factor in helping nearly every participant improve their health trajectory. The findings from this study highlight the potential of story theory to enhance adolescent-friendly HIV care services. Future research should explore how integrating story theory into these services could impact the health outcomes of this population.

## Supporting information

**S1 File. Story paths.**
(PDF)

## Acknowledgments

The authors wish to acknowledge the participants of this study for their contribution of time and invaluable perspectives.

## Author contributions

**Conceptualization:** Derrick Nuwahereza, Charles Peter Osingada.

**Data curation:** Derrick Nuwahereza, Imelda Namatovu.

**Formal analysis:** Charles Peter Osingada.

**Investigation:** Derrick Nuwahereza, Imelda Namatovu.

**Supervision:** Allen Nabisere, Tom Denis Ngabirano.

**Validation:** Allen Nabisere, Imelda Namatovu.

**Writing – original draft:** Charles Peter Osingada.

**Writing – review & editing:** Derrick Nuwahereza, Allen Nabisere, Imelda Namatovu, Tom Denis Ngabirano, Charles Peter Osingada.

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
