## [Decision Letter · Decision Letter 0]

21 Nov 2025

Dear Dr. Osingada,

Thank you for submitting your manuscript to PLOS ONE. After careful consideration, we feel that it has merit but does not fully meet PLOS ONE’s publication criteria as it currently stands. Therefore, we invite you to submit a revised version of the manuscript that addresses the points raised during the review process.

We look forward to receiving your revised manuscript.

Kind regards,

Anthony A. Olashore, PhD.

Academic Editor

PLOS ONE

Journal Requirements:

3. We note that your Data Availability Statement is currently as follows: All relevant data are within the manuscript and its Supporting Information files

4. Please remove all personal information, ensure that the data shared are in accordance with participant consent, and re-upload a fully anonymized data set.

Reviewers' comments:

Reviewer's Responses to Questions

**Comments to the Author**

1. Is the manuscript technically sound, and do the data support the conclusions?

Reviewer #1: Yes

Reviewer #2: Yes

Reviewer #3: Yes

2. Has the statistical analysis been performed appropriately and rigorously?

Reviewer #1: N/A

Reviewer #2: Yes

Reviewer #3: N/A

3. Have the authors made all data underlying the findings in their manuscript fully available?

Reviewer #1: No

Reviewer #2: Yes

Reviewer #3: Yes

4. Is the manuscript presented in an intelligible fashion and written in standard English?

Reviewer #1: Yes

Reviewer #2: Yes

Reviewer #3: Yes

Reviewer #1: There should have to be more context to the lower Troughs in terms of duration of diagnosis but iit is just a suggestion. The article will also have benefited from some more explanatory note on whether doule consent from both the parent or guardian.and from the partcipants below 15,not just the guardian. The article covered important highlights but wil benefit from more descriptive demographics of the environment.

Reviewer #2: The writing is clear, concise, and easy read in standard English. All methodological processes have been presented in their common order. The theoretical framework conjoins well with the methodology, the analysis and discussions. Only few highlighted inconsistences need to be corrected. Also, ethical issues as indicated in the script comments

Reviewer #3: This is very interesting potentially impactful piece of work on HIV/AIDS, the work has great lessons that can be employed in health system provision at large. Briefly explain the story theory.

To further improve the reliability of your research consider the following:

1. Use a checklist (such as the Consolidated criteria for reporting qualitative studies) to systematically report the qualitative findings.

2. Research Team and reflexivity

a. Show experience that researchers have

b. Share reflexivity of the study.

c. How many participants refused to participate and how many dropped off? Any repeat interviews?

3. Study design

a. Briefly show how the story theory was applied in the study

b. How was the tool (? interview guide) piloted?

4. Data Collection, Analysis and findings

a. What did member checking ( if any) show?

b. How many audio recordings were in English and how many were in Luganda?

c. Who translated the Luganda audio recordings, if any?

d. Who read the transcripts?

e. What did field notes show?

f. Show “how many” ( by saying most participants / generally most participants) supported the theme : refer to your discussion section where you did that very well.

**Do you want your identity to be public for this peer review?** For information about this choice, including consent withdrawal, please see our Privacy Policy

Reviewer #1: No

Reviewer #2: **Yes:** Dr. Hlanganiso Roy

Reviewer #3: No

---

## [Author Response · Author response to Decision Letter 1]

1 Dec 2025

Response to comments from Reviewer 1

Thank you for the suggestion to provide more context regarding the lower troughs and duration of diagnosis. This context is already included in the final sentence of the first paragraph of the Introduction, which discusses the challenges faced by young people living with HIV, including their struggle to cope with the diagnosis.

Regarding the issue of double consent, all participants under 18 provided assent. Additionally, as indicated in the ethics section, parents or guardians were requested to provide permission for their child to be involved in the study.

The comment regarding the demographics of the environment was unclear. However, the first sentence of the second paragraph in the Introduction and the first paragraph of the Results section already provide the relevant demographic information for this study.

Response to comments from Reviewer 2

Thank you for the positive feedback on the coherence of the paper and for highlighting the inconsistencies. These inconsistencies have been corrected as indicated on pages 6 and 7, and we have added a statement on page 6 indicating that counseling services were available in case any participant experienced emotional distress during the interview.

Response to comments from Reviewer 3

A brief explanation of the story theory has been included on page 4

The consolidated criteria for reporting qualitative research (COREQ) were used in reporting the findings.

The experience of the researchers has been highlighted on page 7

A reflexivity statement has been included in the first paragraph on page 8.

One participant withdrew from the study. No repeat interviews were conducted, and this has been noted in the Data Collection section on page 6.

A description of how the theory was used to guide the development of the interview guide and interview process is provided on page 6.

Information about how the interview guide was piloted has been included on page 6.

Member checking was not conducted, and this has been acknowledged on page 15.

There were 11 English recordings and 4 Luganda recordings, and this has been noted on page 7 in the Data Management and Analysis section.

The Luganda audios were translated by I.N and A.N who are fluent in English and Luganda.

The transcripts were read by OCP, and this had been made clear on page 7

The field notes were taken but were not included in the analysis.

Thank you for pointing this out and it has been corrected on page 9.

---

## [Decision Letter · Decision Letter 1]

6 Mar 2026

Crests, troughs, and plateaus: Using story theory to explore the experiences of adolescents and young people living with HIV in Kampala, Uganda

PONE-D-25-36890R1

Dear Dr. Osingada,

We’re pleased to inform you that your manuscript has been judged scientifically suitable for publication and will be formally accepted for publication once it meets all outstanding technical requirements.

Kind regards,

Anthony A. Olashore, MBCHB, PhD,

Academic Editor

PLOS One

Additional Editor Comments (optional):

Reviewers' comments:

Reviewer's Responses to Questions

**Comments to the Author**

Reviewer #2: All comments have been addressed

Reviewer #3: All comments have been addressed

Reviewer #4: All comments have been addressed

2. Is the manuscript technically sound, and do the data support the conclusions?

Reviewer #2: Yes

Reviewer #3: Yes

Reviewer #4: Yes

3. Has the statistical analysis been performed appropriately and rigorously?

Reviewer #2: Yes

Reviewer #3: N/A

Reviewer #4: Yes

4. Have the authors made all data underlying the findings in their manuscript fully available?

Reviewer #2: Yes

Reviewer #3: Yes

Reviewer #4: Yes

5. Is the manuscript presented in an intelligible fashion and written in standard English?

Reviewer #2: Yes

Reviewer #3: Yes

Reviewer #4: Yes

Reviewer #2: The authors sufficiently responded to my concerns. More importantly on the availability of the counselling services for those who may have been affected by the interviews.

Reviewer #3: This work is commendable and add to be body of knowledge in caring for adolescents living with HIV/AIDS.

Reviewer #4: All concerns previously expressed have been addressed adequately. The manuscript is well written and easy to read

**Do you want your identity to be public for this peer review?** For information about this choice, including consent withdrawal, please see our Privacy Policy

Reviewer #2: **Yes:** Hlanganiso Roy, Ph.D, Clinical Psychology

Reviewer #3: No

Reviewer #4: **Yes:** Olorunfemi Ogunwobi

---

## [Editor Report · Acceptance letter]

PONE-D-25-36890R1

PLOS One

Dear Dr. Osingada,

I'm pleased to inform you that your manuscript has been deemed suitable for publication in PLOS One. Congratulations! Your manuscript is now being handed over to our production team.

Kind regards,

on behalf of

Dr. Anthony A. Olashore

Academic Editor

PLOS One